# The Effects of β-Hydroxy-β-Methylbutyrate (HMB) on Chemotaxis, Phagocytosis, and Oxidative Burst of Peripheral Blood Granulocytes and Monocytes in Goats

**DOI:** 10.3390/ani9121031

**Published:** 2019-11-26

**Authors:** Roman Wójcik, Katarzyna Ząbek, Joanna Małaczewska, Stanisław Milewski, Edyta Kaczorek-Łukowska

**Affiliations:** 1Department of Microbiology and Clinical Immunology, Faculty of Veterinary Medicine, University of Warmia and Mazury in Olsztyn, Oczapowskiego 13, 10-718 Olsztyn, Poland; joanna.malaczewska@uwm.edu.pl (J.M.); edyta.kaczorek@uwm.edu.pl (E.K.-Ł.); 2Department of Sheep and Goat Breeding, Faculty of Animal Bioengineering, University of Warmia and Mazury in Olsztyn, ul. Oczapowskiego 5, 10-917 Olsztyn, Poland; katarzyna.zabek@uwm.edu.pl (K.Z.); stanislaw.milewski@uwm.edu.pl (S.M.)

**Keywords:** goats, chemotaxis, phagocytosis, oxidative burst, peripheral blood granulocytes and monocytes, β-hydroxy-β-methylbutyrate (HMB)

## Abstract

**Simple Summary:**

The main focus of industrial livestock production is to maximise production output without compromising the well-being of animals, which is why animal diets are supplemented with various feed additives. Feed additives boost immunity and protect animals against pathogens. The list of potential feed additives includes β-hydroxy-β-methylbutyrate (HMB) which occurs naturally in small quantities in citrus fruit, avocado, asparagus, cauliflower, selected fish species, red wine, milk, and alfalfa. However, its mechanism of action and effects on immune system cells have not been thoroughly investigated in animals, including goats. In the present study, the experimental goats whose diets were supplemented with HMB over a period of 60 days were characterised by higher levels of chemotactic and phagocytic activity and a higher rate of oxidative metabolism of peripheral blood granulocytes and monocytes than control group animals whose diets were not supplemented. Granulocytes and monocytes constitute the first line of defence against pathogens and protect animals against disease. They play a particularly important role in young animals which are more susceptible to viral and bacterial infections. Feed additives can deliver numerous benefits by boosting immunity and preventing the spread of infectious diseases in goats.

**Abstract:**

The objective of this study was to determine the effect of β-hydroxy-β-methylbutyrate (HMB) on the chemotactic activity, phagocytic activity, and oxidative metabolism of peripheral blood granulocytes and monocytes in goats. Goat kids aged 30 ± 3 days were divided into two groups of 12 animals each: I—control, and II—experimental. Experimental group animals were fed a diet supplemented with HMB in the amount of 50 mg/Kg BW; whereas the diets of control goats were not supplemented. At the beginning of the experiment (day 0) and on experimental days 15, 30, and 60, blood was sampled from the jugular vein to determine and compare chemotactic activity (MIGRATEST^®^ kit), phagocytic activity (PHAGOTEST^®^ kit), and oxidative metabolism (BURSTTEST^®^ kit) of peripheral blood granulocytes and monocytes by flow cytometry. The analyses of the chemotactic and phagocytic activity of granulocytes and monocytes revealed statistically higher levels of phagocytic activity in the experimental group than in the control group, as expressed by the percentage of phagocytic cells and mean fluorescence intensity. HMB also enhanced the oxidative metabolism of both granulocytes and monocytes, expressed by the rate of oxidative metabolism and mean fluorescence intensity after stimulation with *Escherichia coli* bacteria and PMA (4-phorbol-12-β-myristate-13-acetate).

## 1. Introduction

The implementation of restrictions on antibiotic use in animal production has spurred the search for alternative solutions. The above has increased the popularity of feed additives, including probiotics, prebiotics, and synbiotics [1,2,3], as well as natural and synthetic immunostimulants [4,5,6,7,8] which elicit specific immune responses and boost immunity. Feed additives influence the immune system by enhancing the production of bone marrow stem cells, stimulating immune cells to produce and release cytokines, and by increasing phagocytic activity.

The list of feed additives with immunostimulatory effects includes β-hydroxy-β-methylbutyrate (HMB), an organic hydroxy acid compound and a metabolite of the branched-chain amino acid leucine and alpha-ketoisocaproic acid (α-KIC). In the body, leucine undergoes reversible transamination to α-KIC which can be directly metabolised into HMB in the presence of cytosolic ketoisocaproate dioxygenase. Only around 5% of leucine is metabolised to HMB, and the remainder is transaminated to α-KIC and converted to isovaleryl-CoA by the mitochondrial branched-chain α-keto acid dehydrogenase complex in the liver [9,10,11]. Small quantities of HMB occur naturally in various food products, including citrus fruit, avocado, asparagus, cauliflower, selected fish species, red wine, milk, and alfalfa [12,13]. As an endogenous compound, HMB is produced mainly in muscles and the liver. However, the supply of both endogenous and dietary HMB is often insufficient, and supplementation may be required.

In humans, HMB supplements promote muscle regeneration after physical training and contribute to muscle growth, because HMB not only stimulates the synthesis of proteins, but also prevents their degradation [14,15,16]. In animals, feed supplementation with HMB has been shown to increase weight gains and improve performance. Numerous clinical trials conducted on fish [17,18], birds [10,19], and mammals, including pigs [20,21], cattle [22,23], sheep [8,24], and goats [25], demonstrated that HMB boosts immunity, mostly non-specific immune responses, and increases the animals’ resistance to pathogens. Non-specific (natural, innate) defence mechanisms, including anatomic barriers, antibacterial humoral responses, and cellular mechanisms of antibacterial resistance, constitute the first line of defence against pathogens regardless of previous host-pathogen interactions or the type and structure of the antigen. Unlike specific (acquired) immunity which is not present at birth, non-specific immunity components target all antigens. Receptors that recognise pathogens remain unchanged during an individual’s life and are inherited. However, non-specific immune responses are not stored in an individual’s immunological memory [26].

The effects of feed supplementation with HMB on the immune system of animals remain insufficiently investigated [18,19,20,21,22,23,25,27], in particular in goat kids. In a previous study of kids [25], the authors demonstrated that the addition of HMB to concentrate feed increased body weights, daily gains, growth rate, dimensions of musculus longissimus dorsi (m.l.d.) segments, fat thickness in the loin eye area, and improved selected indicators of non-specific humoral immunity in the blood serum, including lysozyme activity, ceruloplasmin activity, and gamma globulin content.

The objective of this study was to determine the effect of HMB on chemotaxis, phagocytosis, and oxidative burst of peripheral blood granulocytes and monocytes in goats.

## 2. Materials and Methods

### 2.1. Ethical Approval

Animal experiments were carried out in conformance with the Animal Protection Law (Journal of Laws of 24 February 2005, No. 33, item 289) and upon the recommendations of the Animal Ethics Committee of the University of Warmia and Mazury in Olsztyn.

### 2.2. Experimental Design

Goat kids aged 30 ± 3 days were divided into two groups of 12 animals each: I—control, and II—experimental. The two groups were characterised by similar body weights. During the 60-day experiment, the animals from both groups were fed diets with identical composition: Witamilk 2 milk replacer (Wipasz, Olsztyn, Poland), Cielak complementary feed mixture (Wipasz, Olsztyn, Poland), and grass haylage. Witamilk 2 was diluted at 1:10 and administered throughout the entire experiment at a dose of 1.5 L/animal/day, whereas the remaining feedstuffs were provided ad libitum. Beginning on the first day of the study, the experimental group was fed β-hydroxy-β-methylbutyrate (HMB, Metabolic Technologies Inc. Ames, IA, USA) in the form of a complementary feed mixture administered at a dose of 50 mg/Kg BW. The HMB dose was set based on the results reported by Siwicki et al. [18]. The amount of the administered feed and leftovers was monitored throughout the study. The chemical composition of feed and leftovers was determined with standard methods. The results were used to determine nutrient intake in both groups during the entire experiment.

### 2.3. Sample Collection

At the beginning of the study (day 0) and on days 15, 30, and 60 of the experiment, blood was sampled from the jugular vein to determine the chemotactic activity (MIGRATEST^®^ kit) (Glycotope Biotechnology GmbH, Heidelberg, Germany), phagocytic activity (PHAGOTEST^®^ kit) (Orpegen Pharma, Heidelberg, Germany), and oxidative metabolism (BURSTTEST^®^ kit) (Orpegen Pharma, Heidelberg, Germany) of peripheral blood granulocytes and monocytes by flow cytometry.

### 2.4. Determination of the Chemotactic Activity of Blood Granulocytes and Monocytes in Goats with the MIGRATEST^®^ Kit

The assay was performed according to the manufacturer’s specifications. Leukocyte-rich plasma (LRP) isolated from heparinised whole blood by spontaneous sedimentation was used in the experiment. LRP from each sample was placed in two cell culture inserts with pore size of 3.0 μM. One insert was transferred to a well containing the chemoattractant N-formyl-methionyl-leucyl-phenylalanine (fMLP) (Glycotope Biotechnology GmbH, Heidelberg, Germany). The other insert was placed in a buffer solution (Glycotope Biotechnology GmbH, Heidelberg, Germany) without the chemotactic peptide as negative control. Chemotaxis was conducted for 30 min at 37 °C. The cells were stained for 10 min with an antibody reagent (FITC-labelled anti-CD62L) (Glycotope Biotechnology GmbH, Heidelberg, Germany) containing counting beads. Vital DNA dye (Glycotope Biotechnology GmbH, Heidelberg, Germany) was added for 5 min on ice before flow cytometry. The samples were analysed by flow cytometry (BD Biosciences, San Jose, CA, USA) to determine the number of migrated granulocytes and to measure L-selectin shedding of activated cells. The results were presented by calculating the chemotactic index. The chemotactic index was calculated by dividing the number of cells that migrated towards fMLP by the number of cells that migrated in the absence of fMLP [28].

### 2.5. Determination of the Phagocytic Activity of Blood Granulocytes and Monocytes in Goats with the PHAGOTEST^®^ Kit 

All test reagents were prepared in accordance with the manufacturer’s recommendations in the leaflet attached to the product. 100 μL of whole heparinised blood chilled to 0 °C and 20 μL of chilled *Escherichia coli* bacteria (Orpegen Pharma, Heidelberg, Germany) were added to each of the two 5 mL test tubes (blue, Beckman Coulter, Fullerton, CA, USA) (negative control and experimental sample) and were shaken for around 3 s at low speed. The experimental sample was incubated for 10 min at 37 °C, and the negative control was placed in an ice bath at 0 °C. After incubation, 100 μL of the quenching solution (Orpegen Pharma, Heidelberg, Germany) was added to each sample, and the samples were shaken. Three ml of the washing solution (Orpegen Pharma, Heidelberg, Germany) chilled to 0 °C was added, the samples were centrifuged for 5 min at 4 °C (250× *g*), and the supernatant was removed. The rinsing procedure was performed twice, and 2 mL of lysing solution (Orpegen Pharma, Heidelberg, Germany) at room temperature was added to each sample. The samples were shaken and incubated at room temperature for 20 min. The samples were centrifuged for 5 min at 4 °C (250× *g*), and the supernatant was removed. Three ml of the washing solution (Orpegen Pharma, Heidelberg, Germany) chilled to 0 °C was added to each sample, the samples were centrifuged for 5 min at 4 °C (250× *g*), and the supernatant was removed. Two hundred μL of the DNA staining solution (Orpegen Pharma, Heidelberg, Germany) chilled to 0 °C was added, the samples were shaken and incubated for 10 min in an ice bath. Cellular phagocytic activity was determined in a cytometer (BD Biosciences, San Jose, CA, USA) in less than 60 min after the last reagent had been added. The Phagotest (Orpegen Pharma, Heidelberg, Germany) involves fluorescein (FITC)-stained *E. coli* bacteria which are phagocytised by macrophages. Cell nuclei are also stained. The test determines the number of phagocytising cells, granulocytes, and monocytes separately, as well as their phagocytic activity, i.e., the number of bacteria absorbed by a single cell based on fluorescence intensity.

### 2.6. Determination of the Oxidative Metabolism of Blood Granulocytes and Monocytes in Goats with the BURSTTEST^®^ Kit

All test reagents were prepared in accordance with the manufacturer’s recommendations in the leaflet attached to the product. Each analysed sample of whole heparinised blood was divided into four test tubes (blue, Beckman Coulter, Fullerton, CA, USA) of 100 μL each and chilled to 0 °C. Twenty μL of chilled *E. coli* bacteria (Orpegen Pharma, Heidelberg, Germany) was added to the first sample (experimental), 20 μL of the washing solution (Orpegen Pharma, Heidelberg, Germany) was added to the second sample (negative control), 20 μL of fMLP (N-formyl-methionyl-leucyl-phenylalanine) (Orpegen Pharma, Heidelberg, Germany) was added to the third sample (low control), and 20 μL of PMA (4-phorbol-12-β-myristate-13-acetate) (Orpegen Pharma, Heidelberg, Germany) was added to the fourth sample (high control). Test tube contents were stirred and incubated for 10 min at 37 °C (excluding the fMLP (Orpegen Pharma, Heidelberg, Germany) sample which was incubated for 7 min). After incubation, each test tube was supplemented with 20 μL of the substrate solution (Orpegen Pharma, Heidelberg, Germany) and was thoroughly shaken. All samples were incubated for 10 min at 37 °C. After incubation, 2 mL of the lysing solution (Orpegen Pharma, Heidelberg, Germany) at room temperature was added. Test tubes were shaken and incubated at room temperature for 20 min. All samples were centrifuged for 5 min at 4 °C (250× *g*), and the supernatant was removed. All test tubes were rinsed once with 3 ml of the washing solution (Orpegen Pharma, Heidelberg, Germany), centrifuged for 5 min at 4 °C (250× *g*), after which the supernatant was removed. Two hundred μL of the staining solution chilled to 0 °C was added to each sample, test tubes were shaken and incubated for 10 min in an ice bath. The intracellular killing activity of phagocytes was determined in a cytometer (BD Biosciences, San Jose, CA, USA) in less than 30 min after the last reagent had been added. Three activators were used for cell stimulation: *E. coli* bacteria (Orpegen Pharma, Heidelberg, Germany), PMA (Orpegen Pharma, Heidelberg, Germany) as the strong activator, and fMLP (Orpegen Pharma, Heidelberg, Germany) as the weak activator. Dihydrorodamine (123-DHR) was oxidised by mitochondria when H_2_O_2_ was added to induce oxidative stress, and it was converted to cation rhodamine 123 (R123), the fluorescence emitter.

### 2.7. FACS Acquisition and Analysis

Flow cytometry was performed with the FACSCanto II cytometer (BD Biosciences, San Jose, CA, USA). Data were acquired with FACSDiva version 6.1.3 software (BD Biosciences, San Jose, CA, USA) and analysed in FlowJo 10 software (Tree Star, Ashland, OR, USA). The cytometry setup and tracking beads (CST; BD Biosciences, San Jose, CA, USA) were used to initialise the photomultiplier tube (PMT). Unstained control cells and a single stain control for every fluorochrome were prepared and used to establish flow cytometric compensation (Figure 1).

Granulocytes and counting beads were identified in PerCP to side scatter (SSC) scatter and depicted in forward scatter (FSC) to SSC scatter. Data acquisition ended after the acquisition of exactly 2000 events in the region of counting beads. The number of events in the region of granulocytes was then counted, and the number of granulocytes in the control sample was compared with the number of granulocytes in the positive control after stimulation with fMLP. The decrease in L-selectin expression can be measured simultaneously. Downregulation of this cell adhesion molecule correlates directly with the activation of granulocytes under exposure to chemotactic factors. Changes in cell shape precede cell migration and can be measured by analysing the changes in forward scatter signals during flow cytometry (Figure 2).

### 2.8. Statistical Analysis

Numerical results were presented as the arithmetic mean ± SD. The obtained results were processed statistically by two-way ANOVA for orthogonal design. In post-hoc analysis, Dunnett’s test was used to compare day 0 with days 15, 30, and 60 in group II (significance of differences between days: (A) *p* < 0.05; (B) *p* < 0.01; (C) *p* < 0.001; (D) *p* < 0.0001), and Tukey’s test for equal groups to compare group II with group I at each time point (significance of differences between groups: * *p* < 0.05; ** *p* < 0.01; *** *p* < 0.001; **** *p* < 0.0001) with the use of GraphPad Prism 7 software. The significance level has been set to 5% HMB.

## 3. Results

The chemotactic activity of peripheral blood granulocytes was analysed, and the results were used to calculate the chemotactic index (Figure 3 and Figure 4). During the entire experiment, the average value of the chemotactic index was significantly (*p* ≤ 0.05, or *p* ≤ 0.001, or *p* ≤ 0.0001) higher in the experimental group (receiving HMB) than in the control group (not receiving any feed additives), relative to the average baseline value (day 0) of the chemotactic index.

The average oxidative burst activity (metabolism of highly reactive oxygen species) in peripheral blood granulocytes was also higher in the experimental group than in the control group relative to baseline values (day 0) throughout the entire experiment. The average number of cells stimulated to undergo respiratory burst (Figure 5 and Figure 6) and average fluorescence intensity (MFI) in individual granulocytes (Figure 7) were higher in animals whose diets were supplemented with HMB. However, a significant (*p* ≤ 0.05 or *p* ≤ 0.01 or *p* ≤ 0.0001) increase in the average values of the above parameters was noted only in response to stimulation with PMA, a potent activator of protein kinase C, and *E. coli* bacteria during the entire experiment. After stimulation with fMLP, a weak respiratory burst activator, significant (*p* ≤ 0.05) differences in the percentage of granulocytes capable of respiratory burst were observed only on day 15 (Figure 5).

Similar results were noted in peripheral blood monocytes stimulated with *E. coli*, PMA, and fMLP. The average percentage of monocytes undergoing respiratory burst (Figure 8 and Figure 9) and the average oxidative burst intensity (Figure 10) were significantly higher (*p* ≤ 0.05 or *p* ≤ 0.0001) in experimental than in control goats, relative to average baseline values (day 0). When monocytes were stimulated with fMLP, significant (*p* < 0.05) differences in the average percentage of monocytes undergoing oxidative burst (Figure 8) and in the average intensity of oxidative burst in individual monocytes (Figure 10) were observed only on day 15.

The average phagocytic activity of peripheral blood granulocytes was significantly (*p* ≤ 0.05 or *p* ≤ 0.01) higher in goats whose diets were supplemented with HMB than in control group animals, relative to baseline values (day 0) during the entire experiment. However, the above increase was noted only in the average percentage of granulocytes demonstrating phagocytic activity (Figure 11, Figure 12), whereas average fluorescence intensity (MFI), which denotes the number of bacteria phagocytised by a single phagocyte, increased significantly (*p* ≤ 0.05 or *p* ≤ 0.001) only between days 15 and 30 (Figure 13).

Average phagocytic activity (Figure 14 and Figure 15) and average fluorescence intensity (Figure 16) of peripheral blood monocytes were also higher in the experimental group than in the control group, but at different levels of significance (*p* ≤ 0.05 or *p* ≤ 0.01 or *p* ≤ 0.001 or *p* ≤ 0.0001), relative to baseline values (day 0) throughout the entire experiment. 

## 4. Discussion

Non-specific immune responses play a particularly important role in young animals which are more susceptible to viral and bacterial infections than adult individuals. Feed additives with proven immunostimulatory effects can be administered to boost immunity and prevent infectious diseases in many animal species, including goat kids. 

Phagocytosis appears to be a crucial component of non-specific immunity. This process involves professional phagocytes, including polymorphonuclear neutrophils, monocytes, monocyte-derived macrophages and tissue-resident macrophages which rapidly accumulate in the site of infection, as well as non-professional phagocytes, namely the remaining cell types that can perform phagocytosis, such as epithelial cells, fibroblasts, and dendritic cells (DCs). Non-professional phagocytes have a more restricted set of targets, they engulf their targets more slowly, and their phagocytic capacity differs significantly in scope and efficiency from that of professional phagocytes. Phagocytosis is a complex process during which pathogen-associated molecular patterns (PAMPs) are recognised and bound by pattern recognition receptors (PRRs): dectin-1, toll-like receptor 2 (TLR-2), complement receptor 3 (CR3), lactosylceramide (CDw17), class A scavenger receptors (SR-A/CD204,) and macrophage receptors with collagenous structure (MARCO). During phagocytosis, pathogens are internalised and killed by phagocytes. Pattern recognition receptors are not highly specific, and they generally recognise and bind various PAMPs [26,29,30,31,32,33,34,35]. Intercellular killing mechanisms are activated when an engulfed pathogen is trapped inside a phagolysosome, a cytoplasmic body. Non-oxidative intercellular killing mechanisms do not require oxygen, and they activate alkaline proteins and hydrolytic enzymes. Oxidative intercellular killing mechanisms (oxygen burst) produce reactive oxygen species (ROS) [30,34,35,36,37,38,39]. 

In this study, the choice of HMB was dictated by a general absence of clinical data describing its efficacy in goats. The experiment was also undertaken to validate the authors’ previous findings [25]. Recent research has demonstrated that HMB exerts multiple effects, but its mechanism of action has not yet been fully elucidated. The analysed compound influences protein metabolism, in particular in skeletal muscles, as demonstrated in vitro by Ostaszewki et al. [40] and in vivo by Wilkinson et al. [41]. The benefits of HMB in livestock breeding have been described by Van Koevering et al. [42] in feedlot steers, and by Nissen et al. [43], Moore et al. [44] and Qiao et al. [10] in poultry, where the analysed additive improved daily gains, feed intake, feed conversion, carcass traits, and tissue composition. Despite the above, little is known about the immunostimulatory properties of HMB. The ability of the leucine metabolite α-KIC to stimulate the proliferation of ovine lymphocytes in vitro has been described already in 1988 by Kuhlman et al. [27].

According to Townsend et al. [45], in men, dietary supplementation with HMB can decrease the expression of tumour necrosis factor alpha (TNF-α) and TNF-α receptor 1 (TNFR1) on CD14+ monocytes measured by flow cytometry in circulating blood. In a study by Miyake et al. [46], HMB did not influence TE-1 cell proliferation, but it inhibited the production of NF-ĸB and IL-6 on the TE-1 cancer cell line in vitro. Hoffman et al. [47] demonstrated that 23 days of HMB supplementation in combat soldiers can attenuate the inflammatory response (TNF-α, interleukins IL-8 and IL-10, granulocyte colony-stimulating factor (GC-SF), interferon (IFN)-γ, and fractalkine) to intense military training, and maintain muscle quality. These results suggest that the increase in phagocytosis observed in the experimental group in the present study could not be directly linked with HMB’s stimulatory effect on the production of proinflammatory cytokines. It could be postulated that HMB contributes to the phagocytosis of bacteria and other pathogens by enhancing their opsonisation by complement proteins, antibodies or soluble lectin receptors, or by promoting the binding of opsonised pathogens by specialised membrane receptors on the surface of phagocytes. The above hypothesis is confirmed by Gonzales et al. [48] who reported an increase in the expression of complement receptor 3 (CR3) on CD14^+^ monocytes in men orally administered β-hydroxy-β-methylbutyrate free acid (HMB-FA). Complement receptor 3 cannot initiate phagocytosis and plays an auxiliary function by binding molecules responsible for adhesion and the activation of phagocytosis, such as complement regulatory protein iC3b. Józefowski et al. [49] demonstrated that CR3 (β2 integrin CD11b/CD18) cooperates mainly with class A scavenger receptors (SR-A) during phagocytosis. In turn, SR-A are PRRs which act as an “alarm system” by detecting pathogens (Gram-positive and Gram-negative bacteria, viruses, fungi, protozoa) that cross protective barriers in the body. Class A scavenger receptors are capable of binding both endogenous and exogenous ligands on the surface of pathogen cells, which can also stimulate phagocytosis. Similar observations were made in a study by Bowdish et al. [50], where LPS-induced expression of macrophage receptor MARCO, another PRR, enhanced the capture of *Neisseria meningitidis* by phagocytes. MARCO and SR-A have overlapping specificities for various bacterial ligands. Both bind intact bacteria, microbial cell-wall products (lipopolysaccharide, LPS, and lipoteichoic acid, LTA), bacterial CpG DNA and certain modified self-proteins [51,52,53]. According to Peiser et al. [54], Mukhopadhyay et al. [55] and Mukhopadhyay et al. [56], scavenger receptor mediated phagocytosis is not required for pro-inflammatory cytokine production.

In the current study, the chemotactic activity of granulocytes was significantly higher in the group of goats whose diets were supplemented with HMB than in control group animals. There is a general scarcity of published findings on HMB’s effects on this stage of phagocytosis, not only in goats, but also in other animal species; therefore, our results cannot be compared with literature data. Research conducted on other animal species has demonstrated that HMB exerts general stimulatory effects on phagocytosis or stages of phagocytosis other than chemotaxis [5,19,57,58,59]. 

In our experiment, a significant increase in phagocytic activity (expressed by the percentage of phagocytes) and the average number of bacteria eliminated by one phagocyte (expressed by the average fluorescence intensity of monocytes and granulocytes) was observed in the experimental group administered HMB relative to the control group. Similar results were reported in vitro by Peterson et al. [57], where macrophages exposed to HMB demonstrated higher expression of Fc receptors on the surface. The above response could indicate that HMB has two modes of action, or that it directly increases the expression of receptor Fc at the genetic level, or that it improves cell membrane integrity by promoting receptor expression and/or stability on the cell surface. In an in vivo study of rainbow trout (*Oncorhynchus mykiss*), Siwicki et al. [18] found that HMB approximately doubled the potential killing activity ability of polymorphonuclear (PMN) and morphonuclear (MN) cells (*p* < 0.01) relative to the control group. In contrast, an HMB-supplemented diet did not affect the phagocytic potential of abdominal macrophages or the elimination of *Escherichia coli* and *Salmonella arizonae* from the bloodstream of birds [58].

In the present study, HMB also intensified the intracellular killing activity of granulocytes and monocytes stimulated with PMA and *E. coli*, expressed by the percentage of stimulated cells and average fluorescence intensity. Similar observations were made by Siwicki et al. [18] during the respiratory burst assay (RBA) in a spectrophotometric analysis of PMN and MN cells isolated from the head kidney. In their study, the production of strongly oxidising agents increased by nearly 100% during eight weeks of dietary supplementation with HMB. Pronephric phagocytes isolated from rainbow trout (*Oncorhynchus mykiss*) and carp (*Cyprinus carpio*) and grown in a culture medium (RPMI-1640) containing 0, 0.1, 1, 5, 10, 25, 50, or 100 μg of HMB/ml were also far more effective in generating respiratory burst, in particular under exposure to >50 μg of HMB/mL [60]. In a study of young broiler chickens, Peterson et al. [58] noted higher nitrite levels in macrophage culture supernatants in birds receiving HMB than in the control group, despite the fact that HMB did not exert a significant effect on the phagocytic potential of peritoneal macrophages. In an in vitro study, Peterson et al. [57] reported higher nitrite levels in macrophages cultured in the presence of HMB, which could suggest that similarly to other macrophage activators, such as bacterial LPS, HMB, stimulates the activity of nitric oxide synthase (NOS). Kunttu et al. [59] reported that HMB enhanced the microbial killing activity (ROS-response) of leukocytes and increased plasma lysozyme concentrations and complement activity in rainbow trout.

Recent years have witnessed the search for feed additives that not only maximise growth performance, but also boost immunity. The results of this experiment and previous studies suggest that HMB meets the above requirements and could be widely used for immunoprevention in young goats. However, further research is needed to determine HMB’s mode of action and to identify potential differences in its stimulatory effect on immunocompetent cells in various animal species, including monogastric and polygastric animals. 

## 5. Conclusions

This study demonstrated significantly higher levels of granulocyte chemotactic activity (expressed by the chemotactic index) and phagocytic activity (expressed by the percentage of phagocytes), a significantly higher average number of bacteria eliminated by one phagocyte (expressed by the average fluorescence intensity of monocytes and granulocytes), and significantly higher intracellular killing activity of granulocytes and monocytes stimulated with PMA and *E. coli* (expressed by the percentage of stimulated cells and average fluorescence intensity) in experimental goats administered HMB than in control group animals whose diets were not supplemented.

## Figures and Tables

**Figure 1 animals-09-01031-f001:**
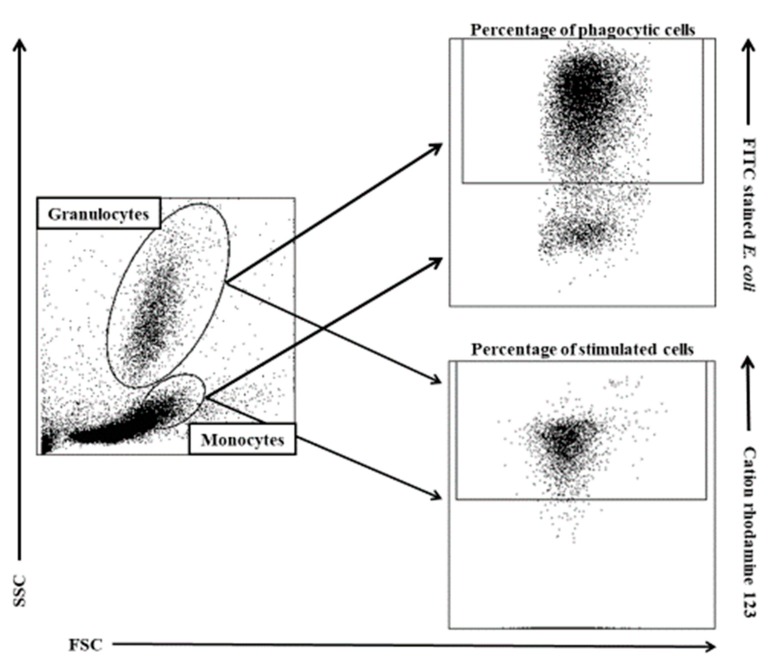
Gating strategy for analysing flow cytometry data. Granulocytes and monocytes were gated based on forward and side scatter (FSC/SSC) parameters. Each cell subset was analysed for the relative number of phagocytising cells and cells stimulated for respiratory burst (fMLP, PMA, or *E. coli* bacteria). fMLP: N-formyl-methionyl-leucyl-phenylalanine, PMA: 4-phorbol-12-β-myristate-13-acetate.

**Figure 2 animals-09-01031-f002:**
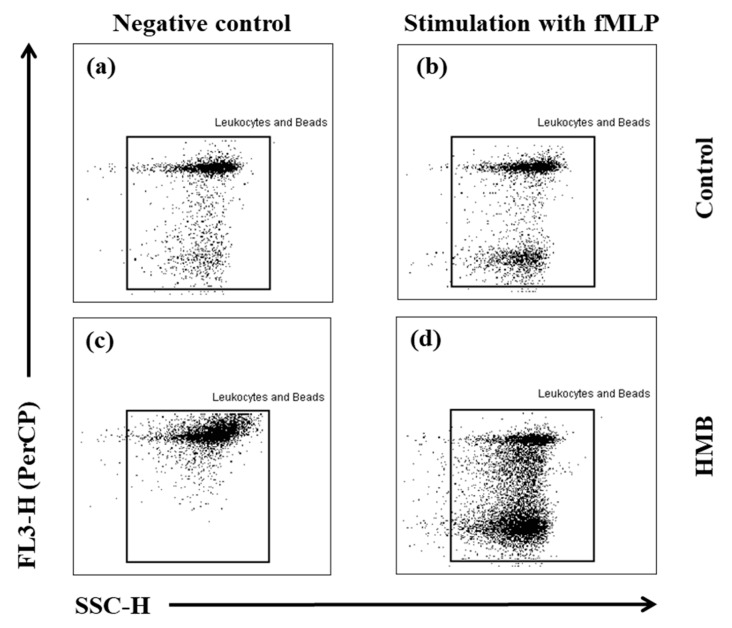
Gating strategy for analysing flow cytometry data based on the granulocyte migration assay. Granulocytes and counting beads were identified in PerCP to SSC scatter and depicted in FSC to SSC scatter: (**a**) control without stimulation; (**b**) control stimulated with fMLP; (**c**) HMB without stimulation; (**d**) HMB stimulated with fMLP.

**Figure 3 animals-09-01031-f003:**
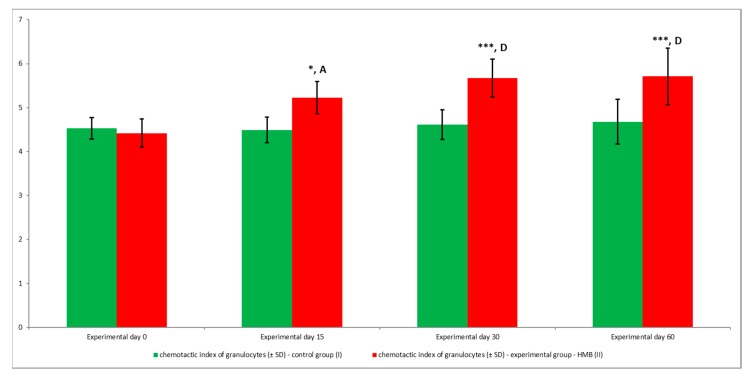
The chemotactic index was calculated by dividing the number of cells that migrated towards fMLP by the number of cells that migrated in the absence of fMLP. Key: I—control group; II—experimental group; SD—standard deviation; Numerical results were presented as the arithmetic mean ± SD. The significance level was set at 0.05. Asterisks refer to statistically significant differences between control and experimental group within the same sampling day at * *p* < 0.05; *** *p* < 0.001; A and D refer to statistically significant differences between day “0” and the consecutive sampling days within experimental group at A: *p* < 0.05; D: *p* < 0.0001.

**Figure 4 animals-09-01031-f004:**
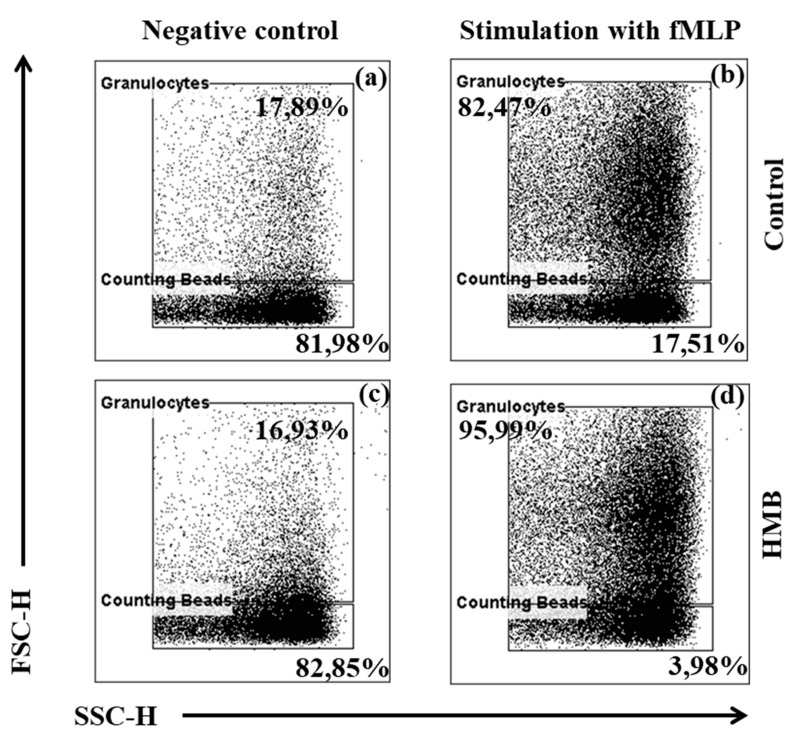
Representative dot plot cytograms showing the percentage of migrating granulocytes and counting beads: (**a**) control without stimulation; (**b**) control stimulated with fMLP; (**c**) HMB without stimulation; (**d**) HMB stimulated with fMLP. The percentage of granulocytes relative to the number of whole cells was determined in each sample as soon as 2000 counting beads were acquired. Stimulation with fMLP increased the number of migrated granulocytes (**b**), (**d**) relative to the controls without fMLP (**a**), (**c**). The chemotactic activity of spontaneously migrating granulocytes and granulocytes migrating towards fMLP increased in the experimental group (supplemented with HMB) relative to the non-supplemented control group.

**Figure 5 animals-09-01031-f005:**
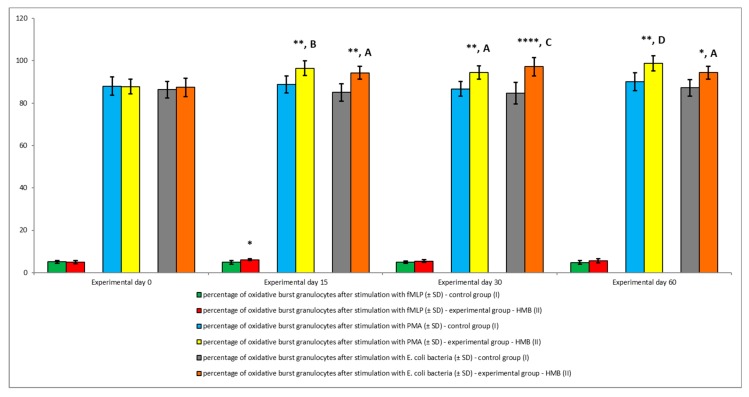
Percentage of granulocytes stimulated to undergo respiratory burst in groups after stimulation with fMLP, PMA, and *E. coli*, as determined in the Bursttest^®^ kit. Key: I—control group; II—experimental group; SD—standard deviation. Numerical results were presented as the arithmetic mean ± SD. The significance level was set at 0.05. Asterisks refer to statistically significant differences between control and experimental group within the same sampling day at * *p* < 0.05; ** *p* < 0.01; **** *p* < 0.0001; A, B, C, D refer to statistically significant differences between day “0” and the consecutive sampling days within experimental group at A: *p* < 0.05; B: *p* < 0.01; C: *p* < 0.001; D: *p* < 0.0001.

**Figure 6 animals-09-01031-f006:**
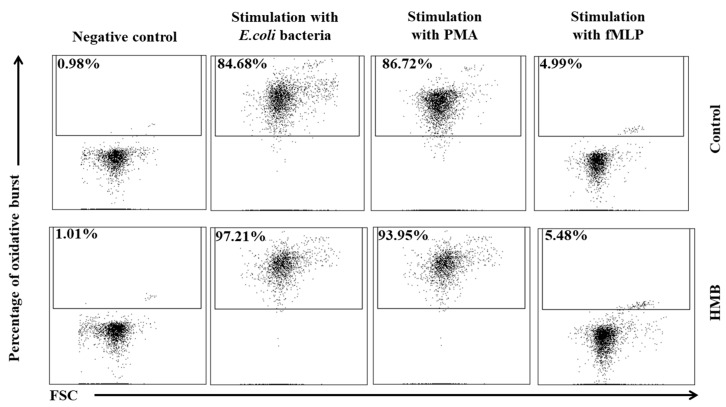
Dot plot cytogram showing the percentage of granulocytes stimulated to undergo respiratory burst in control and experimental goats on experimental day 30. Whole heparinised blood from control and experimental animals (control and HMB) was divided into four test tubes. The samples were combined with the washing solution (negative control), *E. coli* bacteria (opsonising stimulus), PMA (strong stimulus), or fMLP (weak stimulus) and incubated with dihydrorhodamine 123 in a water bath at a temperature of 37 °C. After incubation, cells were lysed and the DNA staining solution was added. The percentages of granulocytes stimulated to undergo respiratory burst (conversion of dihydrorhodamine 123 to rhodamine 123) were gated.

**Figure 7 animals-09-01031-f007:**
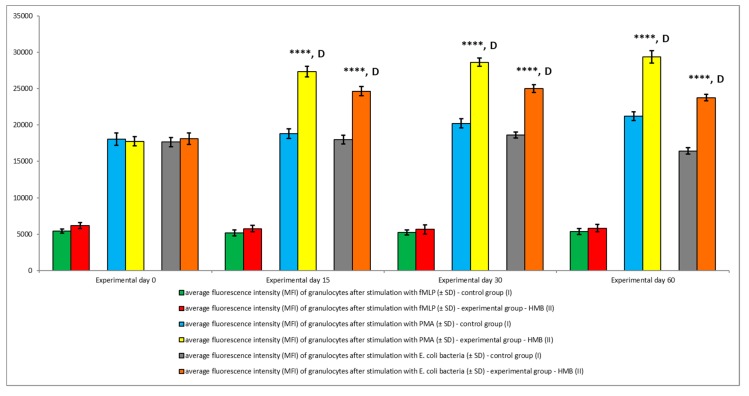
Mean fluorescence intensity (MFI) of granulocytes in groups after stimulation with fMLP, PMA, and *E. coli*, as determined in the Bursttest^®^ kit. Key: I—control group; II—experimental group; SD—standard deviation. Numerical results were presented as the arithmetic mean ± SD. The significance level was set at 0.05. Asterisks refer to statistically significant differences between control and experimental group within the same sampling day at **** *p* < 0.0001; D refer to statistically significant differences between day “0” and the consecutive sampling days within experimental group at D: *p* < 0.0001.

**Figure 8 animals-09-01031-f008:**
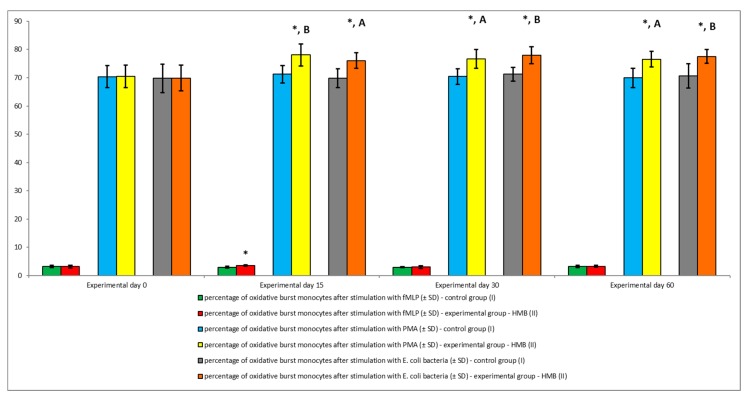
Percentage of monocytes stimulated to undergo respiratory burst in groups after stimulation with fMLP, PMA, and *E. coli*, as determined in the Bursttest^®^ kit. Key: I—control group; II—experimental group; SD—standard deviation. Numerical results were presented as the arithmetic mean ± SD. The significance level was set at 0.05. Asterisks refer to statistically significant differences between control and experimental group within the same sampling day at * *p* < 0.05; A and B refer to statistically significant differences between day “0” and the consecutive sampling days within experimental group at A: *p* < 0.05; B: *p* < 0.01.

**Figure 9 animals-09-01031-f009:**
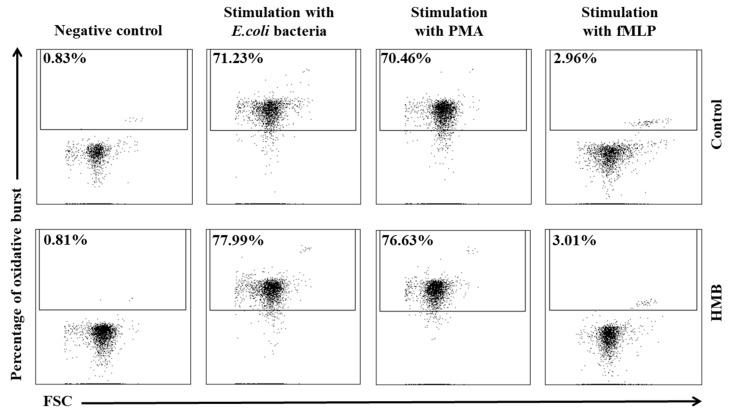
Dot plot cytogram showing the percentage of monocytes stimulated to undergo respiratory burst in control and experimental animals on experimental day 30. The percentages of monocytes stimulated to undergo respiratory burst were gated.

**Figure 10 animals-09-01031-f010:**
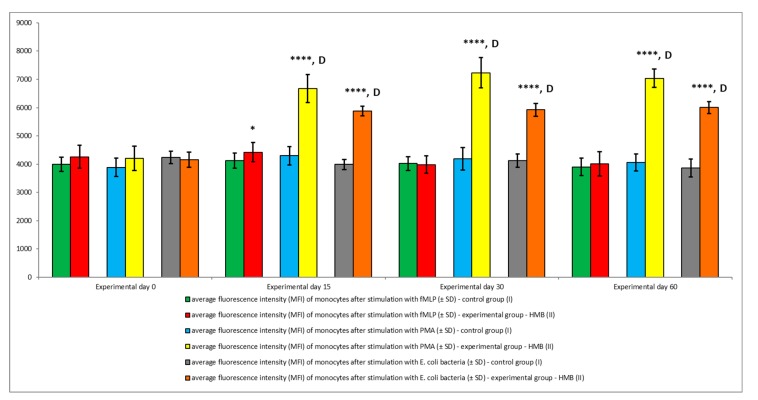
Mean fluorescence intensity (MFI) of monocytes in groups after stimulation with fMLP, PMA, and *E. coli*, as determined in the Bursttest^®^ kit. Key: I—control group; II—experimental group; SD—standard deviation. Numerical results were presented as the arithmetic mean ± SD. The significance level was set at 0.05. Asterisks refer to statistically significant differences between control and experimental group within the same sampling day at **** *p* < 0.0001; D refer to statistically significant differences between day “0” and the consecutive sampling days within experimental group at D: *p* < 0.0001.

**Figure 11 animals-09-01031-f011:**
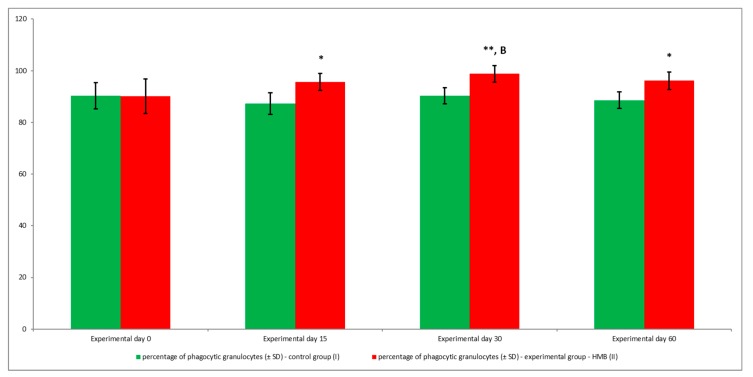
Percentage of phagocytic granulocytes in groups, as determined in the Phagotest^®^ kit. Key: I—control group; II—experimental group; SD—standard deviation. Numerical results were presented as the arithmetic mean ± SD. The significance level was set at 0.05. Asterisks refer to statistically significant differences between control and experimental group within the same sampling day at * *p* < 0.05; ** *p* < 0.01; B refer to statistically significant differences between day “0” and the consecutive sampling days within experimental group at B: *p* < 0.01.

**Figure 12 animals-09-01031-f012:**
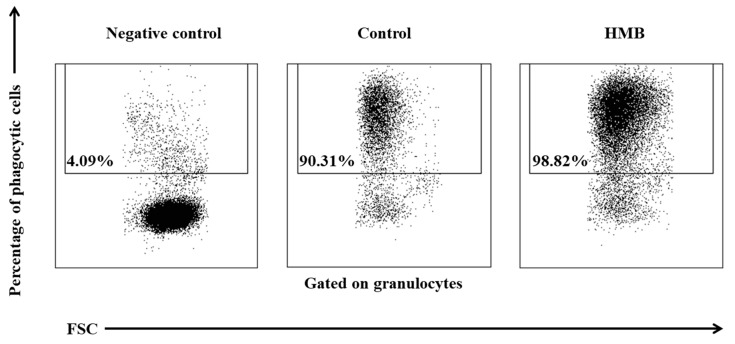
Dot plot cytogram showing the percentage of phagocytic granulocytes in control and experimental goats on experimental day 30. Whole heparinised blood from control and experimental animals was incubated for 10 minutes with an FITC-labelled *E. coli* in an ice bath at a temperature of 0 °C (negative control) or in a water bath at a temperature of 37 °C (control and HMB). The percentages of granulocytes with ingested *E. coli* (FITC) bacteria were gated.

**Figure 13 animals-09-01031-f013:**
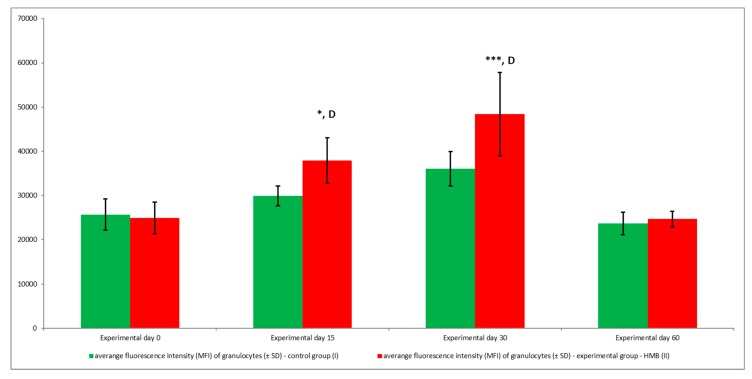
Mean fluorescence intensity (MFI) of granulocytes in groups, as determined in the Phagotest^®^ kit. Key: I—control group; II—experimental group; SD—standard deviation. Numerical results were presented as the arithmetic mean ± SD. The significance level was set at 0.05. Asterisks refer to statistically significant differences between control and experimental group within the same sampling day at * *p* < 0.05; *** *p* < 0.001; D refer to statistically significant differences between day “0” and the consecutive sampling days within experimental group at D: *p* < 0.0001.

**Figure 14 animals-09-01031-f014:**
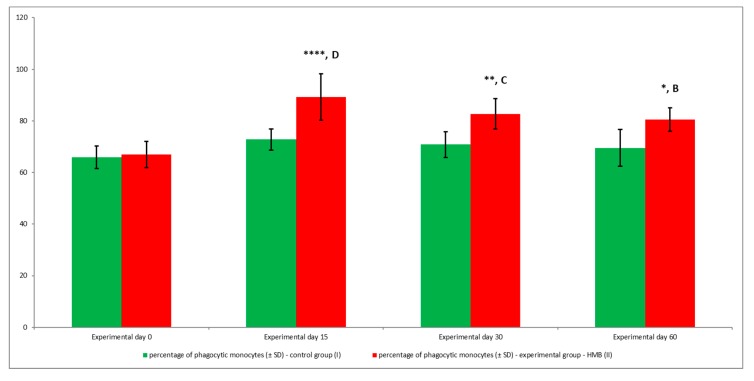
Percentage of phagocytic monocytes in groups, as determined in the Phagotest^®^ kit. Key: I—control group; II—experimental group; SD—standard deviation. Numerical results were presented as the arithmetic mean ± SD. The significance level was set at 0.05. Asterisks refer to statistically significant differences between control and experimental group within the same sampling day at * *p* < 0.05; ** *p* < 0.01; **** *p* < 0.0001; B, C and D refer to statistically significant differences between day “0” and the consecutive sampling days within experimental group at B: *p* < 0.01; C: *p* < 0.001; D: *p* < 0.0001.

**Figure 15 animals-09-01031-f015:**
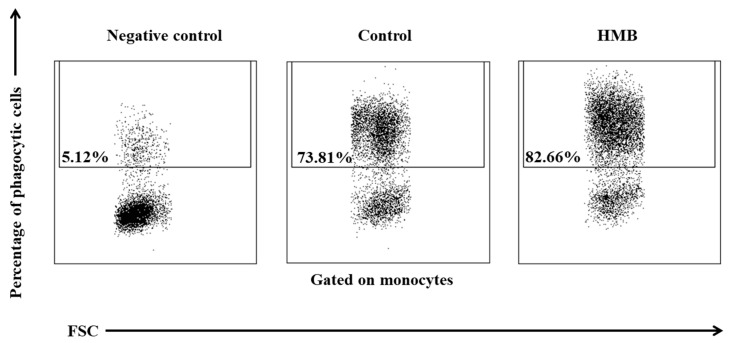
Dot plot cytogram showing the percentage of phagocytic monocytes in control and experimental goats on experimental day 30. Whole heparinised blood from control and experimental animals was incubated for 10 minutes with FITC-labelled *E. coli* in an ice bath at a temperature of 0 °C (negative control) or in a water bath at a temperature of 37 °C (control and HMB). The percentages of monocytes with ingested *E. coli* (FITC) bacteria were gated.

**Figure 16 animals-09-01031-f016:**
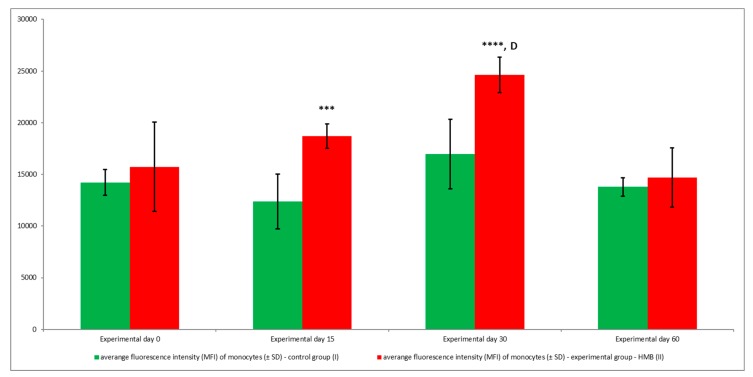
Mean fluorescence intensity (MFI) of monocytes in groups, as determined in the Phagotest^®^ kit. Key: I—control group; II—experimental group; SD—standard deviation. Numerical results were presented as the arithmetic mean ± SD. The significance level was set at 0.05. Asterisks refer to statistically significant differences between control and experimental group within the same sampling day at *** *p* < 0.001; **** *p* < 0.0001; D refer to statistically significant differences between day “0” and the consecutive sampling days within experimental group at D: *p* < 0.0001.

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
