# Peer review of "The Effects of β-Hydroxy-β-Methylbutyrate (HMB) on Chemotaxis, Phagocytosis, and Oxidative Burst of Peripheral Blood Granulocytes and Monocytes in Goats"

_animals, 2019, doi:10.3390/ani9121031_

Round 1
Reviewer 1 Report
Comments to the author: The authors presented an interesting study aimed to show the was to determine the effect of β-hydroxy-β-methylbutyrate (HMB) on chemotaxis, phagocytosis, and oxidative burst of peripheral blood granulocytes and monocytes in goats.
The structure of the manuscript appears adequate and well divided in the sub-paragraphs. The methodology is well described with enough experimental data and results to support the work.
The study is easy to follow and covers an interesting topic, but some minor issues should be improved before publication.
Several typos should be corrected thorough the text. English must be improved.
Discussion Section: This paragraph required a general revision for an improving and to eliminate redundant sentences and to add some some “take home message”.
Author Response
We would like to thank the Reviewer for a thorough perusal of our paper. We greatly appreciate the Reviewer’s constructive comments which helped us improve the manuscript. All of the Reviewer’s concerns have been addressed and the manuscript has been revised accordingly, except for one suggestion:
Discussion section: add some “take home message” – In our opinion, the last paragraph of the Discussion section and the subsequent Conclusions section can be regarded as a take home message.Reviewer 2 Report
Wojcik and colleagues evaluate the effect of hydroxyl-methylbutyrate supplementation on neutrophil biology in livestock. This includes chemotaxis, phagocytic activity as well as oxidative metabolism. The manuscript is generally well written and the reported findings may improve livestock production by preventing viral and bacterial infections. The experimental design is fair, well controlled and the analysis well performed as well. All the claims make by the authors are supported by the presented results. Therefore, I do not have any major concern to do not recommend this manuscript for publication. Nevertheless, there are a number of minor comments authors should address.
1- The major criticism is that it is not clear to me if the authors used granulocytes, neutrophils or monocytes. They use granulocytes and neutrophils as interchangeable terms (eg line 284 and then legend in figure 11), when neutrophils are a type of granulocytes. Also, neutrophils account for only a proportion of white cells in perypheral blood. Did the authors purify these cells before performing the chemotaxis/phagocytosis/oxidative metabolism experiments? If not, how can the authors be sure that they are stimulating (and monitoring) these cells and not others? This last question is related to the shown cytograms in which I do not see separation using cell surface markers specific for neutrophils or monocytes.
2- Hydroxy methyl butyrate is a metabolite produced by the gut microbiota. Because the role played by these microorganisms in stimulating immune system, I wonder whether the authors could elaborate on this in the discussion.
3- It would be great, if the authors combine related figures into panels.
4- As mentioned before, the manuscript is well written, but I recommend proof-reading by a native speaker. In some instances, I am not sure if some words are properly used. For example line 95: “… feedstuffs were fed ad libitum.” “… feedstuffs were provided ad libitum”. Also latin expressions such as ad libitum should be in italics.
5- Please revise the units. In some instances liter is referred as L (upper case) and microliter as µl (lower case). The same applies for Kg, which should be in upper case.
Author Response
We would like to thank the Reviewer for a thorough perusal of our paper. We greatly appreciate the Reviewer’s constructive comments which helped us improve the manuscript. The Reviewer’s concerns have been addressed as follows:
As regards the first comment, please note that the tests used in the study are commercial tests in which the isolated and evaluated cells are referred to as granulocytes and monocytes, which is justified following their gating in the FSS/SSC system. We agree that the term “neutrophils”, denoting one of granulocyte subpopulations, is not fully justified in this context because specific surface markers were not used for the above cells. Therefore, the general term “granulocytes” should be used to describe this group of cells, which is why “neutrophils” have been replaced with “granulocytes” in the revised manuscript. Both the gut microbiota and their role in stimulating the immune system are topical and interesting issues, but they are also very extensive. Therefore, if they were addressed in the Discussion, the length of this section would considerably increase and the readers could miss its main points. Please note that the combining of all related figures into panels would lead to the accumulation of large amounts of numbers, letters and symbols presenting the results of tests and statistical analyses, written using a tiny font (so as to fit all data in one panel). In our opinion, such a solution could reduce the readability of graphically presented data. As recommended by the Reviewer, the manuscript has been edited by a professional translator. It has also been spell-checked and grammar-checked by a native English speaker. The relevant corrections have been made.